# Dosimetry of a Novel ^111^Indium-Labeled Anti-P-Cadherin Monoclonal Antibody (FF-21101) in Non-Human Primates

**DOI:** 10.3390/cancers15184532

**Published:** 2023-09-13

**Authors:** Gregory Ravizzini, William Erwin, Louis De Palatis, Lucia Martiniova, Vivek Subbiah, Vincenzo Paolillo, Jennifer Mitchell, Asa P. McCoy, Jose Gonzalez, Osama Mawlawi

**Affiliations:** 1Department of Nuclear Medicine, University of Texas MD Anderson Cancer Center, 1400 Pressler St., Unit 1483, Houston, TX 77030, USAjagonzalez1@mdanderson.org (J.G.); 2Department of Imaging Physics, Division of Diagnostic Imaging, University of Texas MD Anderson Cancer Center, Houston, TX 77030, USA; werwin@mdanderson.org (W.E.); omawlawi@mdanderson.org (O.M.); 3Technology and Business Development, Center for Advanced Biomedical Imaging, University of Texas MD Anderson Cancer Center, Houston, TX 77030, USA; ldp.biodev@gmail.com; 4Department of Experimental Therapeutics, University of Texas MD Anderson Cancer Center, Houston, TX 77030, USA; lmartini@mdanderson.org; 5Department of Investigational Cancer Therapeutics, Division of Cancer Medicine, University of Texas MD Anderson Cancer Center, Houston, TX 77030, USA; 6Cyclotron Radiochemistry Facility, Center for Advanced Biomedical Imaging, University of Texas MD Anderson Cancer Center, Houston, TX 77030, USA; vincenzo.paolillo@mdanderson.org; 7Department of Veterinary Medicine and Surgery, University of Texas MD Anderson Cancer Center, Houston, TX 77030, USA; jmitchell2@mdanderson.org

**Keywords:** *CDH3*, SPECT, non-human primates, radioimmunotherapy, internal radiation dosimetry, theranostics

## Abstract

**Simple Summary:**

This research study focuses on the P-cadherin protein, which is found in many types of tumors, and could be a potential target for therapy. FF-21101, a human–mouse chimeric monoclonal antibody that targets P-cadherin, was labeled with indium-111(^111^In) and investigated for biodistribution and radiation doses in six cynomolgus macaques. To assess the radiation profile distribution and clearance of FF-21101(^111^In), we performed whole-body imaging and generated time–activity curves. The lungs, spleen, liver, kidneys, and heart wall received the highest radiation dose estimates for FF-21101(^111^In). These findings led us to estimate the radiation doses that would be delivered to different organs if this antibody was labeled with yttrium-90 (^90^Y) and used in targeted radioimmunotherapy in patients. FF-21101 shows potential for clinical application in humans, according to biodistribution data obtained from macaques. Based on these results, an investigational new drug application was filed with the FDA for a Phase I clinical trial.

**Abstract:**

P-cadherin is associated with a wide range of tumor types, making it an attractive therapeutic target. FF-21101 is a human–mouse chimeric monoclonal antibody (mAb) directed against human P-cadherin, which has been radioconjugated with indium-111 (^111^In) utilizing a DOTA chelator. We investigated the biodistribution of FF-21101(^111^In) in cynomolgus macaques and extrapolated the results to estimate internal radiation doses of ^111^In- and yttrium-90 (^90^Y)-FF-21101 for targeted radioimmunotherapy in humans. Whole-body planar and SPECT imaging were performed at 0, 2, 24, 48, 72, 96, and 120 h post-injection, using a dual-head gamma camera. Volumes of interest of identifiable source organs of radioactivity were defined on aligned reference CT and serial SPECT images. Organs with the highest estimated dose values (mSv/MBq) for FF-21101(^111^In) were the lungs (0.840), spleen (0.816), liver (0.751), kidneys (0.629), and heart wall (0.451); and for FF-21101(^90^Y) dose values were: lungs (10.49), spleen (8.21), kidneys (5.92), liver (5.46), and heart wall (2.61). FF-21101(^111^In) exhibits favorable biodistribution in cynomolgus macaques and estimated human dosimetric characteristics. Data obtained in this study were used to support the filing of an investigational new drug application with the FDA for a Phase I clinical trial.

## 1. Introduction

Cadherins are cell–cell adhesion glycoproteins that form calcium-dependent intercellular junctions and play an essential role in morphogenesis, development, and maintenance of the integrity of tissues and organs [1]. The cadherin family is subdivided into various subfamilies, including the classical E-, P-, and N-cadherins, each demonstrating a specific tissue distribution [2]. E-cadherin is expressed in all epithelial tissues; however, the expression of P-cadherin is restricted to the basal or inner layers of stratified epithelia, including prostate and skin, and to breast myoepithelial cells [3]. Several studies have demonstrated that aberrant P-cadherin expression is associated with cell proliferation and with tumors of the colon, breast, lung, thyroid, pancreas, and cervix, making P-cadherin (encoded by the gene *CDH3*) an excellent tumor-associated target [4,5].

The development of P-cadherin anticancer therapy commenced with PF-03732010, a humanized anti-CDH3 monoclonal antibody (mAb) with demonstrated antitumor activity confirmed by carcinoma mouse models [6]. More recently, a clinical trial (ClinicalTrials.gov accession no. NCT00557505) revealed that this antibody was well tolerated in humans but demonstrated no significant anti-tumor activity in solid tumors to warrant further development as a naked antibody [7]. Nonetheless, due to the favorable toxicity profile of the anti-CDH3 antibody and the interesting features of the target molecule, a radioimmunotherapy approach was proposed to achieve enhanced anti-tumor activity. The method of conjugating targeting antibodies or peptides with radionuclides has been successful in treating well-differentiated neuroendocrine tumors, prostate cancer, and other cancers [8,9,10]. Since high accumulation and homogeneous distribution at the target site and rapid elimination from non-target tissues are desirable for radiopharmaceutical therapy (RPT), the half-life of the radionuclide should be long enough to allow the radiolabeled monoclonal antibody (or peptide) to reach the tumor, and short enough to limit radiation exposure to normal tissues [11]. In addition, the specificity of the selected antibody to the cellular target enables increased therapeutic efficacy while minimizing toxicity to normal tissues [12,13,14,15].

FF-21101, a human–mouse chimeric immunoglobulin G1 (IgG1) monoclonal antibody (mAb) targeting human P-cadherin is being developed as a radio-immuno-therapeutic (RIT) for the treatment of patients with advanced solid tumors. In this paper, we describe FF-21101 as the mAb conjugate to the polydentate macrocyclic ligand, 1,4,7,10-tetraazacyclododecane-1,4,7,10-tetraacetic acid (DOTA) for use in dosimetry, imaging (e.g., with ^111^In), or as a radioimmunotherapeutic (e.g, with ^90^Y). The DOTA chelator forms stable metal complexes with a number of divalent and trivalent metals, including the lanthanide, indium-111 (^111^In) [16].

Tumor geometry is considered to be a significant variable for the success of RIT, and it has been proposed that the emission characteristics of the radionuclide should be matched appropriately to the lesion size/volume to be treated in order to focus energy within the tumor rather than in the surrounding tissues [17]. Therefore, for our study, FF-21101 was labeled with the theranostic pair, ^111^In and ^90^Y. The nuclear properties of ^111^In (physical half-life of 67.3 h and decay via electron capture with gamma emissions of 172 and 245 keV to stable cadmium-111) are well suited for multi-day imaging studies using planar gamma camera imaging and single photon emission computed tomography (SPECT). These nuclear properties are well matched with the 64.1 h physical half-life of yttrium-90 (^90^Y) and the biological half-life of the IgG form of the FF-21101 product [18]. ^90^Y emits beta particles, which penetrate soft tissue to a relatively long spatial depth of 5.3 mm. Therefore, ^90^Y-labeled mAbs cause DNA damage to the targeted cells as well as to adjacent or more distant cells due to their long range [19]. As a result of this cross-fire effect, there is an expectation that FF-21101(^90^Y) will have a more positive therapeutic impact on treating solid tumors than anti-CDH3 therapies alone, making ^90^Y ideal for this purpose [20]. ^90^Y is also used clinically for radioembolization of hepatocellular carcinoma and metastatic disease to the liver from colon cancer using microspheres [21]. Unfortunately, while ^90^Y has a favorable characteristic as a therapeutic radionuclide, the lack of gamma emission makes it suboptimal for imaging and dosimetry and dictates the need to delineate its biodistribution with a γ-emitting surrogate [22,23]. ^111^In has, therefore, been the surrogate radionuclide of choice for ^90^Y for gamma camera imaging due to the similarities in metabolic handling [24] and coordination chemistry with ^90^Y [25,26].

In this study, we evaluated the biodistribution and dosimetry of FF-21101(^111^In) in cynomolgus macaques. We estimated human adult male and female organ radiation absorbed doses by computing non-human primate (NHP) residence times based on serial gamma camera imaging and mass-scaled them to estimate the human equivalents. This was followed by the calculation of human absorbed dose estimates according to the methodology developed by the Medical Internal Radiation Dose (MIRD) Committee [18]. In order to compute internal radiation dose estimates for FF-21101(^90^Y) for radioimmunotherapy, ^111^In residence times obtained from FF-21101(^111^In) were converted to ^90^Y equivalents.

## 2. Material and Methods

### 2.1. Radiolabeling Preparation of FF-21101(^111^In)

The monoclonal antibody (FF-21101) conjugated to DOTA was provided as a 5 mg/mL solution in 250 mM sodium acetate, pH 5.5, by FUJIFILM Diosynth Biotechnologies, USA, Inc., Morrisville, NC, USA. Indium-111 chloride (^111^InCl_3_) was purchased from Mallinckrodt, Inc., St. Louis, MO, USA, and stored securely according to manufacturer specifications until ready for use. A preset volume of FF-21101 was transferred to a sterile 10 mL reaction vial (Greer Laboratories, Lenoir, NC, USA) using aseptic techniques and mixed with ^111^InCl_3_ in a shielded laminar air flow hood; the specific radioactivity of the mixture was 37 MBq/mg at the time of injection. The radioactive dose volume at the time of preparation was calculated based on the radioactive concentration of ^111^InCl_3_ at the time of preparation and added to the reaction vial. The reaction mixture was incubated at 40 °C in a thermomixer for 20 min. Following incubation, clinical formulation buffer (25 mmol/mL sodium acetate, 2 mg/mL ascorbic acid, 0.3 mg/mL diethylene triamine pentaacetic acid, and 0.72% saline; AMRI, Burlington, MA, USA) was added to bring the total volume to 10 mL. Approximately 111 MBq of FF-21101(^111^In) in 5 mL was drawn from the reaction vial into a pre-labeled 10 mL syringe, assayed in a dose calibrator, and delivered for injection. Labeled antibody was purified using a Biospin Column 6 (Bio-Rad, Tokyo, Japan) according to the manufacturer’s instructions. The labeling processes did not result in degradation of these antibodies.

### 2.2. Determination of Specific Activity

The specific activity of FF-21101(^111^In) was calculated based on measurements of radioactivity concentration and molar concentration in the dosing injectate. The radioactivity concentration of FF-21101(^111^In) was determined using a Packard 5500 gamma spectrometer (Perkin-Elmer, Waltham, MA, USA).

### 2.3. Experimental Animals

After a five-week quarantine/acclimation period, six adult (3 male and 3 female) cynomolgus macaques were housed separately in a facility accredited by the Association for Assessment and Accreditation of Laboratory Animal Care International. All experiments were performed following published guidelines under a protocol approved by the Institutional Animal Care and Use Committee. Prior to imaging, the animals were fasted overnight but had free access to water. The animals were anesthetized with an intramuscular injection of ketamine (10–15 mg/kg) plus atropine sulfate (0.04 mg/kg) to facilitate tracheal intubation. Anesthesia was maintained via inhalation of isoflurane (1% to 3%) and oxygen delivered by a Fabius Tiro gas anesthesia system (Drager Medical, Inc., Tokyo, Japan). Cannulas were placed in both saphenous veins of each animal: one cannula was used for administration of the FF-21101(^111^In) and the other was used for repetitive blood sampling. Body temperature was maintained at 37 ± 0.8 °C using an air-circulating heating device (model 505; Arizant Healthcare Inc., Eden Prairie, MN, USA). The electrocardiogram, pulse oximetry, and respiration rates were continuously monitored using an Infinity Gamma XL Monitor (Drager Medical, Inc.). The stage of menstrual cycle in females was not monitored. The use of a vacuum immobilization system bag was instrumental in helping to maintain NHP position throughout each planar/SPECT and subsequent CT scan imaging sessions over several days.

### 2.4. Gamma Camera and SPECT System

All animal whole-body (WB) planar and SPECT gamma camera imaging studies were performed using a Siemens Symbia S scanner (Siemens Medical Solutions USA, Hoffman Estates, IL, USA) equipped with two 3/8″ NaI detectors and medium-energy low-penetration collimators. The reported NEMA ^111^In system planar sensitivity for that system is 430 cpm/µCi.

### 2.5. Gamma Camera and SPECT Imaging Protocol

Each primate was first placed in a vacuum immobilization bag to maintain consistent positioning during each planar/SPECT and CT imaging session. All primates were positioned supine/feet first into the gantry. Approximately 111 MBq of ^111^In-FF-21101 was administered to each primate as a continuous infusion over 10 min. A (0 h) planar/SPECT scanning session commenced immediately after the end of the infusion and consisted of a head-to-foot whole-body planar sweep followed by either a single-bed (female) or two-bed (male), head–neck–trunk SPECT scan. This planar-plus-SPECT-imaging paradigm was repeated multiple times according to the following schedule: 2, 24, 48, 72, 96, and 120 h from the end of infusion. A 10 mL liquid vial ^111^In calibration source (approximately 1.85 MBq) was acquired along with the NHP at each time point, to convert reconstructed SPECT counts to activity.

For whole-body planar imaging, the acquisition parameters were set as follows: a 256 × 1024 acquisition matrix, and a scan speed of 10 cm/min (0, 2, 24 h), 7 cm/min (48, 72 h), or 5 cm/min (96, 120 h), with 15% (±7.5%) energy window widths centered over the two ^111^In photopeaks (172 and 245 keV). For SPECT imaging, the acquisition parameters were: 128 views over 360° (64 views/head and 180° total gantry rotation), 15 sec/view (0, 2, 24 h), 21 sec/view (48, 72 h), or 30 sec/view (96, 120 h). Images were acquired and reconstructed in a 128×128 matrix with a pixel size of 4.8 mm. In addition to the two photopeak images, images from a scatter window below the 245 keV photopeak (211 keV/11%) and scatter windows below and above the 172 keV photopeak (144 keV/18% and 190 keV/7%) were also acquired, for dual- and triple-energy-window scatter compensation, respectively. An ordered-subset expectation-maximum iterative algorithm (8 iterations, 16 subsets, and 9.6 mm full-width at half-maximum Gaussian post-filter), incorporating CT-based attenuation correction (AC) and energy-window-based scatter and system spatial resolution compensations, was employed for reconstruction of the SPECT images. 

### 2.6. CT Imaging

The CT imaging was performed on a GE 750 HD scanner (General Electric Healthcare, Waukesha, WI, USA) and was used for SPECT attenuation correction and to help identify the contours of the source organ’s volumes of interest (VOIs). Primates were positioned centrally on the CT scanner table. A scout scan (AP, 120 kVp, 80 mA, 100 cm, 10 cm/s) was first acquired to determine animal positioning and anatomical coverage. This was followed by a CT scan with the following parameters: 120 kVp, Smart mA tube current modulation, 0.5 s rotation, 64 × 0.625 mm nominal beam width at isocenter, and 1.375 pitch. The AC CT-filtered back-projection reconstruction parameters were 5 mm slice thickness and spacing, 50 cm diameter, and STANDARD filter; while those for the organ-contouring CT were 3.75 mm slice thickness, 3.27 mm slice spacing, 50 cm diameter, and SOFT filter.

### 2.7. Image Analysis for Dosimetry Calculations

Three-dimensional VOIs of identifiable source organs of radioactivity were manually defined on the first-day CT scan and all serial SPECT scans were manually registered to that CT scan using MIM version 6.5 (MIM Software Inc., Beachwood, OH, USA). These VOIs were used for both CT-based organ mass estimation and SPECT organ activity versus time (time–activity curve, [TAC]) generation. The identifiable source organs were heart contents, lungs, liver, spleen, kidneys, and testes. Two-dimensional total-body regions of interest were defined in an analogous fashion using the planar whole-body scan images, for total-body TAC generation.

### 2.8. Blood Sampling

Measurement of radioactivity concentrations in the blood was based on 0.5 mL venous blood samples obtained via the catheterized vein at 1, 2, 5, 10, 15, 20, and 30 min, and 2, 24, 48, 72, 96, and 120 h after FF-21101(^111^In) injection. Approximately 50 mg of each blood sample was assayed along with an ^111^In counting standard, to derive radioactivity concentration versus time using a gamma counter (Cobra Quantum, Perkin-Elmer, Waltham, MA, USA), expressed as a Fraction of Injected Activity per mL (FIA/mL).

### 2.9. Residence Time and Absorbed Dose Calculations

Human organ radiation absorbed dose estimates for FF-21101(^111^In) (mGy/MBq) were obtained using the MIRD methodology. The MIRD equation for estimating mGy/MBq to a target from one or more sources is
$$D_{t}=\sum_{i=1}^{n}T_{s}\times S_{i}(t\leftarrow s)$$
where *D_t_* is the dose to a target (mGy/MBq); *T_s_* is the residence time (or normalized cumulative activity, equal to the total number of disintegrations per unit of administered activity, in h) in source *s*; and *S_i_*(*t* ← *s*) is the average absorbed dose in target *t* per unit of cumulative activity, $\tilde{A_{s}}$, in source *s* (dose factor, in mGy/MBq-h). Cumulative activity and residence time *T_s_* are related as follows:$$\tilde{A_{s}}=A_{0}\times T,\;\mathrm{or}\;\mathrm{equivalently},\;T_{s}=\tilde{A_{s}}/A_{0}$$
where *A*_0_ is administered activity. Residence time was computed based on the imaging data acquired from all macaques as the fraction of administered activity in the source, *f_s_*(*t*), integrated from time *t* = 0 to ∞. The dose factors employed in the dose calculations were those defined in the FDA-cleared and *de facto* industry-standard program OLINDA/EXM 1.1 (Vanderbilt University, Nashville, TN, USA). Human radiation doses were calculated from all cynomolgus macaques and the normalized number of disintegrations using OLINDA/EXM 1.1 for the adult male and female phantoms. Appendix A describe the dosimetry assumptions in detail.

The calculated absorbed doses for FF-21101(^111^In) and FF-21101(^90^Y) were then compared with the published absorbed doses for ibritumomab tiuxetan (Zevalin; Acrotech Biopharma, LLC, East Windsor, NJ, USA) labeled with the same radionuclides (^111^In and ^90^Y) [18].

## 3. Results

### 3.1. FF-21101(^111^In) Synthesis

Radiolabeling of FF-21101 with ^111^InCl_3_ was carried out by the clinical site radiopharmacy. The prescribed radiolabeled activities were assayed using a pre-calibrated dose calibrator, prepared in the empty reaction vial, drawn into an appropriately sized sterile syringe, and sterile-filtered prior to infusion into the subject. Radiochemical purity of the product was assayed using radio-thin layer radiochromatography and typically exceeded 98% up to 24 h post-compounding. Each primate was administered 103.6–189.8 MBq of FF-21101(^111^In) in a volume of 5 mL. FF-21101 antibody concentration administered to each animal ranged from 0.48 to 0.54 mg/mL. Antibody dose ranged from 0.29 to 0.63 mg/kg (mean of 0.34 mg/kg for males and mean of 0.59 mg/kg for females). Simultaneously, with the FF-21101(^111^In) injection, up to 15 mL of saline was infused using a second syringe through the same catheter port; this infusion of saline was continued for ~30 s after the end of the FF-21101(^111^In) administration to steadily flush the infusion line.

### 3.2. Planar/SPECT Imaging of FF-21101(^111^In)

FF-21101(^111^In) accumulated in the spleen, lungs, liver, kidneys, and heart contents in all six primates (Figure 1A). It is noteworthy that the testes of the male NHPs were considered a source organ due to FF-21101(^111^In)-specific accumulation being visible in the whole-body planar images. This necessitated a second bed position SPECT scan over the pelvic region to properly account for the radioactivity in the testes (Figure 1B). Due to the size of male NHPs, the axial extent of the first bed position SPECT scan only covered the head through to the mid-lower abdomen. In contrast, the ovaries of female NHPs did not show radiotracer accumulation above that of the surrounding body tissue, and, hence, were not considered as a source organ. As a result, only a single-bed position SPECT scan was required (Figure 1C). In addition, we have included a representative image of a female from the Phase-I clinical trial with an adenocarcinoma of the vagina, in which tumors surround the urinary bladder (Figure 1D). As a reference, subsequent FDG(^18^F) PET imaging obtained as standard of care demonstrates increased radiotracer uptake localizing to the vaginal tumor that corresponded to the area of FF-21101(^111^In) uptake (Figure 1E).

### 3.3. Pharmacokinetics of FF-21101(^111^In)

The 0 to 120 h planar/SPECT imaging demonstrated that the hepatobiliary system was the primary route for clearance of FF-21101(^111^In). Hence, despite accumulation in the kidneys, radiotracer activity in the urinary bladder activity was not prominent.

The time-dependent biologic FF-21101(^111^In)-derived NHP FIA/mL in whole blood (Figure 2A) reached a mean peak value of 0.0037 ± 0.00093 standard error of the mean (SEM) within 2 h. Radioactivity counts decreased 40% mono-exponentially to a mean of 0.0022 ± 0.00042 SEM within 24 h after injection. At the end of the measurements, blood FIA/mL decreased by an average of 78% to a mean of 0.00079 ± 0.00016 SEM at 120 h. The corresponding time-dependent biologic FF-21101(^111^In)-derived humanized FIA/mL in whole blood are presented in Figure 2B, following the same pattern of decrease in radioactivity.

As expected, relatively low accumulation of FF-21101(^111^In) was observed in most organs not involved in the biological clearance of FF-21101(^111^In)-derived radioactivity (Figure 3). We did not observe retention or accumulation of FF-21101(^111^In) in the brain.

### 3.4. Radiation Dosimetry

Source organ residence times for FF-21101(^111^In) and FF-21101(^90^Y) are shown in Table 1 and Table 2, respectively; the corresponding organ doses are shown in Table 3 and Table 4.

The median biologic half-life for the circulating antibody in whole blood of NHPs was 50.19 h with a median effective half-life of 28.75 h for FF-21101(^111^In). The corresponding median effective half-life for FF-21101(^90^Y) in NHPs was 28.15 h. In comparison, the derived humanized median biologic half-life of FF-21101(^111^In) was 55.69 h with an effective half-life of 29.8 h for FF-21101(^90^Y) (Figure 2C).

Although OLINDA/EXM 1.1 computes the equivalent dose (mSv/MBq), the radiation weighting factors for all the emissions (photon and electron) from ^111^In and ^90^Y are in unity, thus, the equivalent dose is the same as the radiation absorbed dose for both, i.e., 1 mSv = 1 mGy. The results are tabulated for a reference adult human (male and female). In each case, a comparison with the published absorbed doses for ibritumomab tiuxetan labeled with the same radionuclide is also provided [18]. The organs with the five-highest estimated absorbed doses for FF-21101(^111^In) were the lungs (0.840 mSv/MBq), spleen (0.816 mSv/MBq), liver (0.751 mSv/MBq), kidneys (0.629 mSv/MBq), and heart wall (0.451 mSv/MBq). The organs with the five-highest estimated absorbed doses for FF-21101(^90^Y) were similar to those for FF-21101(^111^In): lungs (10.49 mSv/MBq), spleen (8.21 mSv/MBq), kidneys (5.92 mSv/MBq), liver (5.46 mSv/MBq), and heart wall (2.61 mSv/MBq). Comparison between the male and female dose estimates shows similar organ doses, as well as the ^111^In effective dose, suggesting similar biodistribution profiles for the two genders. Similar trends were observed in human dosimetry for ^111^In/^90^Y-ibritumomab tiuxetan (Table 3 and Table 4). For all NHPs, the humanized estimates for FF-21101(^90^Y)-administered activity were below the applicable limits for external beam radiotherapy (<3 Gy to red marrow, <30 Gy to the liver, and <20 Gy to the kidneys and lungs, <1 Gy to the testes, <2 Gy to the ovaries) [27,28].

## 4. Discussion

Radioimmunotherapy and peptide receptor radionuclide therapy are both unique approaches for the targeting and treatment of tumors with a high degree of specificity. Both approaches offer great promise to deliver a lethal dose of radiation to cancer cells while significantly reducing the radiation dose to normal organs. Compounds that have been shown to have such favorable therapeutic abilities and safety profiles are ^90^Y-ibritumomab tiuxetan (as an example for radioimmunotherapy) and ^177^Lu-DOTATATE (Lutathera, Advanced Accelerator Application, USA, Inc., Millburn, NJ, USA, as an example of peptide receptor radionuclide therapy) [29]. FF-21101 is a human–mouse chimeric immunoglobulin of the IgG1 subclass that targets human P-cadherin and may prove to be another successful therapeutic agent for radioimmunotherapy applications.

This study provides information on non-target uptake levels of FF-21101(^111^In/^90^Y) in normal organs and was conducted to facilitate a Phase I clinical trial with FF-21101 radiolabeled with ^111^In for dosimetry and ^90^Y for therapy. All animals tolerated the imaging procedures and intravenous infusion of FF-21101(^111^In) well, with minimal discomfort. The use of the immobilizer was instrumental and helped in the fusion alignment of the planar (WB) and SPECT images, and, thus, decreased the possibility of significant errors due to organ volume contouring or dose calculation errors because of image misregistration. No significant weight loss was observed in the animals. Biodistribution and radiation dosimetry were evaluated after intravenous infusion of FF-21101(^111^In) in six cynomolgus macaques, with an activity range of 103.6–189.8 MBq. The radiation absorbed doses were increased in organs affected by blood flow. The source organs identified by FF-21101(^111^In) in decreasing order by dose were the lungs, spleen, liver, kidneys, and heart wall; and for FF-21101(^90^Y), the lungs, spleen, kidneys, liver, and heart wall. The amount of FF-21101(^111^In) that was associated with the ovaries and testes was similar, given that most of its dose came from surrounding source organs; however, since the testes are in fact source organs (whereas ovaries are not), the dose absorbed by testes from FF-21101(^90^Y) is much higher (1.86 in testes vs. 0.306 mSv/MBq in ovaries), as ^90^Y is a pure beta emitter. Thus, it is important to evaluate the dosimetry profile of animal models in both genders.

The FDA package insert specifies an effective half-life of 30 h for ^90^Y-ibritumomab tiuxetan, which is in agreement with our calculations [30]. The pattern of biodistribution and clearance of FF-21101(^111^In) in NHPs was similar to that reported in humans for ^111^In-ibritumomab tiuxetan [18]. All source organs exhibited uptake and mono-exponential clearance typical of an intact mAb, with the five-highest absorbed dose estimates (D) (mSv/MBq) of FF-21101(^111^In) in the lungs, spleen, liver, kidneys, and heart wall. The order of organs with the five-highest absorbed D values in the ^111^In-ibritumomab tiuxetan studies was slightly different, with the highest exposure to the liver, spleen, kidneys, osteogenic cells, and red marrow. The radiation absorbed dose from ^90^Y-ibritumomab tiuxetan in the lungs was low in comparison to FF-21101(^90^Y) (0.761 and 10.495 mSv/MBq, respectively). The spleen was shown to have the highest absorbed radiation dose from ^90^Y-ibritumomab tiuxetan, yet the dose from FF-21101(^90^Y) was double that of the spleen. The doses to the kidneys and liver followed a similar pattern to that of the spleen. The dose from FF-21101(^90^Y) in red marrow was significantly lower than that of ^90^Y-ibritumomab tiuxetan (0.857 vs. 2.73 mSv/MBq, respectively). The difference in absorbed D values is not unexpected given the mechanism of action of ibritumomab (anti-CD20) versus FF-21101 (anti-P-cadherin). ^111^In-ibritumomab tiuxetan is an immunoconjugate of a murine anti-CD20 mAb approved in 2002 by the FDA for assessing its biodistribution in low-grade or follicular B-cell non-Hodgkin’s lymphoma patients prior to ^90^Y-ibritumomab tiuxetan (Zevalin^®^) therapy [31].

In RPT, tumor cell killing is achieved by delivering ionizing radiation [32]. In our case, we used ^90^Y for the therapeutic component. This radionuclide emits high-energy beta particles (2300 keV maximum beta energy (E_βmax_)), with 90% of its energy being absorbed within a sphere corresponding to 100–200 cell diameters. This gives ^90^Y a broad crossfire effect locally, which is ideal for treating solid tumors [20]. The favorable imaging and dosimetry results of this study further support the possibility of exploring the integration of FF-21101 in future clinical trials. One such Phase I study has already been conducted at the MD Anderson Cancer Center (ClinicalTrials.gov Identifier: NCT02454010) to investigate a number of parameters, including the determination of radiation dose estimates prior to radioimmunotherapy, its human biodistribution properties, and dosimetry profiles in primary solid tumors and metastases [33].

Table 5 compares humanized NHP organ radiation absorbed dose estimates for FF-21101(^90^Y) to reported human-normalized FF-21101(^90^Y) organ radiation absorbed dose estimates based on FF-21101(^111^I) dosimetric imaging reported by Subbiah et al. from a Phase I clinical trial [33]. The NHP values in this table are reported in mGy/mCi to match the units of the published human dose estimates. The table shows that most organ dose estimates correlate relatively well within acceptable uncertainties. The difference between NHP and human values for kidneys, liver, red marrow, and osteogenic cells were all within less than 30% of each other. It was previously reported that the degree of uncertainty associated with the dosimetry data in human and preclinical studies was about 20–47% [34,35,36]. The biggest discrepancies in dose estimates are noted for the spleen, testes, and lungs. Several reasons could be attributed to these discrepancies. For example, the difference in the estimated spleen dose could be attributed to the differences in the subject population between the two studies. For the patient studies, human data were obtained from 16 patients (7 male and 9 female) with *advanced* solid tumors, while the data from the NHP were acquired from 6 *healthy* subjects. Subbiah et al. [33] reported P-cadherin expression in tumors and normal tissues and demonstrated a moderate amount of P-cadherin expression in normal splenic tissue; however, as previously reported by others, nonspecific binding of mAbs as well as macrophage uptake [37,38] may cause increased tracer accumulation in the spleen, in addition to possible aggregation/complexation, which, when combined with patient tumor burden, may explain the 70% higher spleen human dose estimates. The difference in testicular values between the NHP and the human study, on the other hand, could be due to the size of cynomolgus macaques’ testes in comparison to human testes. To delineate the ROIs in NHP, testes present several challenges due to their small size and resultant coarse images that lead to increased partial volume effects, thereby potentially causing the observed difference in dose estimates. Finally, the difference in observed lung dose estimates between the NHP and human study (129%) could be attributed to the effects of placing the primate in a vacuum immobilization bag to maintain consistent positioning during each planar/SPECT- and CT-imaging session. The tracheal intubation and administration of anesthesia could cause the increase in radiotracer accumulation in the lungs due to potential atelectasis of the lung.

## 5. Conclusions

Overexpression of P-cadherin in several types of tumors provides an attractive target for RIT [4,5]. The biodistribution and radiation dosimetry characteristics of FF-21101(^111^In), as determined in NHPs, indicate that the radiation dose of this radiotracer to potential human patients is similar to that of other mAb-based agents used for planar and SPECT imaging (e.g., ibritumomab tiuxetan). The highest radiation dose is delivered to the lungs, followed by the spleen, liver, kidneys, and heart wall but all are within acceptable ranges. However, in this study, we assess that NHP internal dosimetry underestimated the radiation dose estimated in humans by 70% in the spleen and testes and overestimated the dose estimates in the lungs. Yet, the comparison of results in other organs between NHP and human data may be justified, despite differences in the acquisition parameters in tumor-bearing vs. healthy NHP subjects. Taken together, the results of this study confirm that FF-21101 conjugated to ^111^In for imaging and to ^90^Y for radioimmunotherapy can be safely translated to clinical trials.

## Figures and Tables

**Figure 1 cancers-15-04532-f001:**
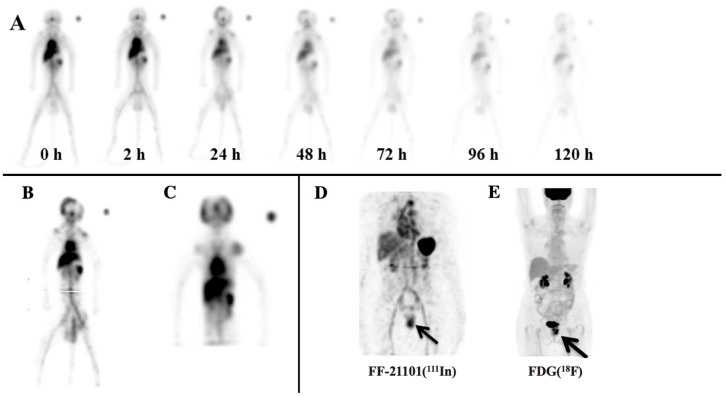
Representative NHP FF-21101(^111^In) scans: (**A**) Male NHP anterior whole-body planar images (female biodistribution was the same, aside from the testes); (**B**) summed coronal view of the 24 h, two-bed SPECT of the same NHP male; and (**C**) summed coronal view of a female 24 h, single-bed SPECT. The scans demonstrate uptake in blood pool, lungs, liver, spleen, kidneys, and testes. (**D**) FF-21101(^111^In) MIP SPECT image of a 69-year-old female with an adenocarcinoma of the vagina. A black arrow indicates that FF-21101(^111^In) activity is increased in the recurrent tumor in the mid-to-lower left vaginal wall. (**E**) Subsequent MIP FDG(^18^F) PET demonstrated high uptake within both urinary bladder and the uterine tumor, thus confirming the location of the adenocarcinoma of the vagina.

**Figure 2 cancers-15-04532-f002:**
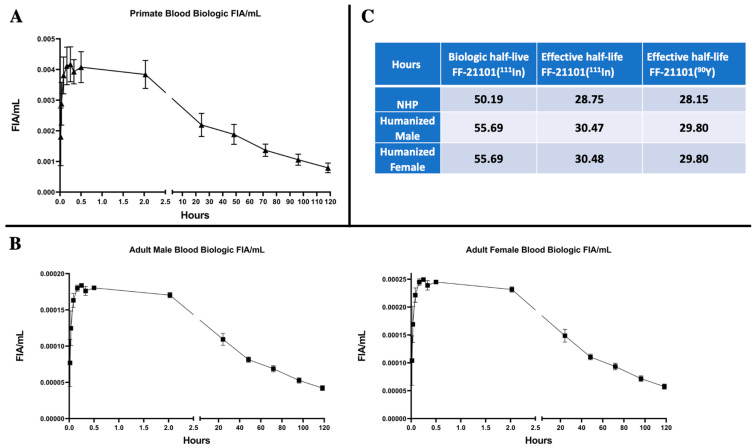
Primate (**A**) and Humanized (**B**) Biologic (decay-corrected) fraction of injected activity per mL (FIA/mL) of FF-21101(^111^In) in the blood. Data points: Mean ± SEM. Biologic and effective half-life (**C**) calculated from FIAs.

**Figure 3 cancers-15-04532-f003:**
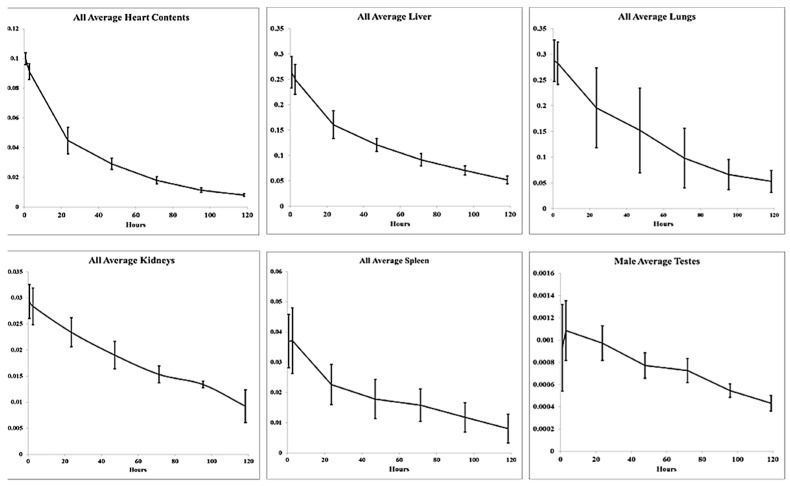
Humanized fraction of FF-21101(^111^In) versus time in identified source organs. Data points: mean ± SD.

**Table 1 cancers-15-04532-t001:** Humanized FF-21101(^111^In) source organ residence times (h) for the derived from NHP dosimetric data.

	FF-21101(^111^In) Residence Time (h)				^111^In-Ibritumomab Tiuxetan Human			
Source Organ	Overall		Females		Males		Published: [18]	
	Mean	Std Dev	Mean	Std Dev	Mean	Std Dev	Mean	Std Dev
**Heart Contents**	**3.845**	0.302	**3.828**	0.342	**3.857**	0.351	**1.18**	0.557
**Kidneys**	**3.412**	0.566	**3.556**	1.095	**3.316**	0.082	**1.4**	0.351
**Liver**	**16.414**	2.194	**15.914**	0.529	**16.747**	3.012	**12.9**	4.92
**Lungs**	**15.407**	2.792	**13.53**	1.471	**16.658**	2.938	**1.41**	0.904
**Spleen**	**3.101**	0.89	**2.256**	0.19	**3.664**	0.613	**1.63**	0.814
**Red Marrow**	**3.085**	0.837	**3.995**	0.103	**2.477**	0.114	**11.7**	4.52
**Testes**	**0.151**	0.019	NA	NA	**0.151**	0.019	**0.163**	0.065
**Total Body**	**84.828**	2.905	**82.191**	2.627	**86.585**	1.357	**82.183**	2.062
**Remainder of Body**	**39.473**	5.467	**39.112**	3.109	**39.714**	7.398	**51.8**	16.2

**Table 2 cancers-15-04532-t002:** Humanized FF-21101(^90^Y) source organ residence times (h) derived from those for FF-21101(^111^In).

	FF-21101(^90^Y) Residence Time (h)					
Source Organ	Overall		Females		Males	
	Mean	Std Dev	Mean	Std Dev	Mean	Std Dev
**Heart Contents**	**3.768**	0.293	**3.654**	0.286	**3.781**	0.340
**Kidneys**	**3.209**	0.494	**3.319**	0.678	**3.128**	0.069
**Liver**	**15.874**	2.109	**16.518**	2.036	**16.219**	2.890
**Lungs**	**14.98**	2.686	**18.603**	9.426	**16.171**	2.839
**Spleen**	**2.972**	0.838	**1.843**	0.573	**3.510**	0.550
**Red Marrow**	**3.028**	0.818	**4.300**	0.665	**2.434**	0.110
**Testes**	**0.142**	0.018	**NA**	NA	**0.142**	0.018
**Total Body**	**81.253**	2.689	**80.591**	3.533	**82.885**	1.247
**Remainder of Body**	**37.339**	5.237	**32.354**	8.500	**37.499**	7.071

**Table 3 cancers-15-04532-t003:** OLINDA/EXM 1.1 organ radiation absorbed dose estimates for FF-21101(^111^In).

							^111^In-Ibritumomab Tiexetan Human	
	Female		Male				Published [18]	
	Average	Std Dev	Average	Std Dev	Average	Std Dev	Average	Std Dev
Organ	mSv/MBq	mSv/MBq	mSv/MBq	mSv/MBq	mSv/MBq	mSv/MBq	mSv/MBq	mSv/MBq
**Adrenals**	**0.310**	0.028	**0.251**	0.011	**0.281**	0.037	**0.202**	0.041
**Brain**	**0.067**	0.013	**0.059**	0.010	**0.063**	0.011	**0.082**	0.020
**Breasts**	**0.136**	0.018	**0.113**	0.003	**0.124**	0.017	**0.080**	0.016
**Gallbladder Wall**	**0.317**	0.019	**0.268**	0.020	**0.293**	0.032	**0.229**	0.059
**LLI Wall**	**0.105**	0.018	**0.090**	0.013	**0.098**	0.017	**0.130**	0.027
**Small Intestine**	**0.130**	0.014	**0.118**	0.010	**0.124**	0.013	**0.145**	0.027
**Stomach Wall**	**0.189**	0.004	**0.164**	0.005	**0.176**	0.014	**0.139**	0.028
**ULI Wall**	**0.152**	0.014	**0.128**	0.008	**0.140**	0.016	**0.147**	0.027
**Heart Wall**	**0.498**	0.046	**0.403**	0.025	**0.451**	0.061	**0.195**	0.048
**Kidneys**	**0.671**	0.114	**0.586**	0.008	**0.629**	0.086	**0.315**	0.071
**Liver**	**0.851**	0.112	**0.650**	0.100	**0.751**	0.145	**0.467**	0.202
**Lungs**	**0.999**	0.445	**0.681**	0.101	**0.840**	0.337	**0.152**	0.036
**Muscle**	**0.124**	0.004	**0.104**	0.005	**0.114**	0.012	**0.103**	0.021
**Ovaries**	**0.112**	0.018	NA	NA	**0.112**	0.018	**0.138**	0.028
**Pancreas**	**0.290**	0.009	**0.257**	0.012	**0.274**	0.020	**0.199**	0.041
**Red Marrow**	**0.181**	0.012	**0.145**	0.005	**0.163**	0.022	**0.272**	0.065
**Osteogenic Cells**	**0.276**	0.012	**0.218**	0.019	**0.247**	0.035	**0.312**	0.050
**Skin**	**0.068**	0.005	**0.060**	0.005	**0.064**	0.006	**0.065**	0.015
**Spleen**	**0.675**	0.150	**0.958**	0.140	**0.816**	0.202	**0.464**	0.193
**Thymus**	**0.197**	0.022	**0.167**	0.012	**0.182**	0.023	**0.114**	0.024
**Thyroid**	**0.095**	0.003	**0.141**	0.038	**0.118**	0.035	**0.098**	0.024
**Urinary Bladder Wall**	**0.089**	0.018	**0.087**	0.010	**0.088**	0.013	**0.113**	0.028
**Uterus**	**0.107**	0.019	NA	NA	**0.107**	0.019	**0.132**	0.029
**Testes**	NA	NA	**0.114**	0.044	**0.114**	0.044	**0.175**	0.076
**Total Body**	**0.162**	0.009	**0.133**	0.002	**0.148**	0.017	**0.124**	0.022
**Effective Dose Eq**	**0.382**	0.054	**0.339**	0.023	**0.360**	0.044		
**Effective Dose**	**0.279**	0.050	**0.241**	0.015	**0.260**	0.039		

**Table 4 cancers-15-04532-t004:** OLINDA/EXM 1.1 organ radiation absorbed dose estimates for FF-21101(^90^Y).

							^90^Y-Ibritumomab Tiuxetan Human	
	Female		Male				Published [18]	
	Average	Std Dev	Average	Std Dev	Average	Std Dev	Average	Std Dev
Organ	mSv/MBq	mSv/MBq	mSv/MBq	mSv/MBq	mSv/MBq	mSv/MBq	mSv/MBq	mSv/MBq
**Adrenals**	**0.306**	0.081	**0.274**	0.052	**0.290**	0.063	**0.379**	0.118
**Brain**	**0.306**	0.081	**0.274**	0.052	**0.290**	0.063	**0.379**	0.118
**Breasts**	**0.306**	0.081	**0.274**	0.052	**0.290**	0.063	**0.379**	0.118
**Gallbladder Wall**	**0.306**	0.081	**0.274**	0.052	**0.290**	0.063	**0.379**	0.118
**LLI Wall**	**0.306**	0.081	**0.274**	0.052	**0.290**	0.063	**0.379**	0.118
**Small Intestine**	**0.306**	0.081	**0.274**	0.052	**0.290**	0.063	**0.379**	0.118
**Stomach Wall**	**0.306**	0.081	**0.274**	0.052	**0.290**	0.063	**0.379**	0.118
**ULI Wall**	**0.306**	0.081	**0.274**	0.052	**0.290**	0.063	**0.379**	0.118
**Heart Wall**	**2.707**	0.231	**2.513**	0.190	**2.610**	0.217	**1.080**	0.394
**Kidneys**	**6.333**	1.296	**5.497**	0.119	**5.915**	0.942	**2.440**	0.610
**Liver**	**6.353**	0.785	**4.573**	0.817	**5.463**	1.210	**3.640**	1.389
**Lungs**	**12.277**	6.206	**8.710**	1.528	**10.493**	4.489	**0.761**	0.488
**Muscle**	**0.306**	0.081	**0.274**	0.052	**0.290**	0.063	**0.379**	0.118
**Ovaries**	**0.306**	0.081	**NA**	NA	**0.306**	0.081	**0.379**	0.118
**Pancreas**	**0.306**	0.081	**0.274**	0.052	**0.290**	0.063	**0.379**	0.118
**Red Marrow**	**1.008**	0.071	**0.705**	0.053	**0.857**	0.175	**2.730**	0.909
**Osteogenic Cells**	**1.283**	0.023	**0.742**	0.087	**1.013**	0.302	**2.140**	0.519
**Skin**	**0.306**	0.081	**0.274**	0.052	**0.290**	0.063	**0.379**	0.118
**Spleen**	**6.41**	1.989	**10.010**	1.552	**8.210**	2.536	**4.650**	2.327
**Thymus**	**0.306**	0.081	**0.274**	0.052	**0.290**	0.063	**0.379**	0.118
**Thyroid**	**0.306**	0.081	**0.274**	0.052	**0.290**	0.063	**0.379**	0.118
**Urinary Bladder Wall**	**0.306**	0.081	**0.274**	0.052	**0.290**	0.063	**0.379**	0.118
**Uterus**	**0.306**	0.081	**NA**	NA	**0.306**	0.081	**0.379**	0.118
**Testes**	**NA**	NA	**1.860**	0.233	**1.860**	0.233	**2.160**	0.842
**Total Body**	**0.738**	0.035	**0.588**	0.010	**0.663**	0.086	**0.587**	0.107

**Table 5 cancers-15-04532-t005:** Humanized OLINDA/EXM 1.1 organ radiation absorbed dose estimates for FF-21101(^90^Y) compared to reported human-normalized FF-21101(^90^Y) organ radiation absorbed dose estimates based on FF-21101(^111^I) dosimetric imaging.

Organ	Human Average (mGy/mCi) *	Std Dev	Primate Average (mGy/mCi) **	Std Dev	% Change
**Spleen**	**1031.5**	399.9	**303.77**	93.83	70.55
**Testes**	**339.5**	122.6	**68.82**	8.62	79.73
**Kidneys**	**219.7**	75.8	**218.86**	34.85	0.38
**Lungs**	**168.9**	56.9	**388.24**	166.09	129.86
**Liver**	**160.6**	58.5	**202.13**	44.77	25.86
**Heart Wall**	**89.7**	18.1	**96.57**	8.03	7.66
**Osteogenic Cells**	**34.7**	13.2	**37.48**	11.17	8.01
**Red Marrow**	**29.6**	11	**31.71**	6.48	7.13
**Total Body**	**25.7**	2.6	**24.53**	3.18	4.55
**Ovaries**	**14.6**	4.3	**11.32**	3.00	22.45
**Uterus**	**14.6**	4.3	**11.32**	3.00	22.45
**All other organs**	**12.5**	5.1	**10.73**	2.33	14.16

* Normalized average values of total dose estimates from 16 patients (7 male, 9 female) with advanced solid tumors refractory to or relapsed from prior therapy, as reported in Subbiah et al. [33]. ** Humanized average (mean) organ absorbed dose estimated and normalized from 6 healthy non-human primates (3 male, 3 female), reported as mSv/MBq in Table 4, and recalculated to mGy/mCi for comparison purposes.

## Data Availability

The data presented in this study is available upon request from the corresponding author.

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
