# Peer review of "Dosimetry of a Novel ^111^Indium-Labeled Anti-P-Cadherin Monoclonal Antibody (FF-21101) in Non-Human Primates"

_cancers, 2023, doi:10.3390/cancers15184532_

Round 1
Reviewer 1 Report
In this work, the mAB FF-21101 was labeled with In-111 and imaging data of non-human primates were exptrapolatet to project normal organ absorbed doses in human beings. The dosimetry of In-111/Y-90 Zevalin is provided as standard of reference. The quality of this reaserch is high and the paper is well written.
However, the FF-21101 mAB labeled with In-111 and Y-90 was already studied in a first phase-1 trial (Ref. 33). Now the dosimetry estimates in Non-human primates are reported with some delay, which is reducing novelity and clinical relevance.
From the academical perspective I would find it very interesting to see, how the data of this study, i.e. dosimetry extrapolation of In-111 FF-21101 from primates to human beings, compare to the dosimetry of In-111 FF-21101 directly done with human beings. In other world the own results should be compared to the results of Ref.-33 in a dedicated table! This would address the very interesting question how reliable it really is to approxmiate dosimetry of human beings by using non-human primates.
Reviewer 2 Report
The Authors investigated the the biodistribution of FF-21101(111In), a human-mouse chimeric monoclonal antibody (mAb) directed against human P-cadherin and radioconjugated with indium-111 (111In) utilizing a DOTA chelator, in cynomolgus macaques. They also extrapolated the results to estimate internal radiation doses of 111In- and yttrium-90 (90Y)-FF-21101 for targeted radioimmunotherapy in humans. The topic is relevant, and the article is of interest. Methods are scientifically sound and results are clear.
I have some advice for the discussion section. Since data on humans has been recently published (Subbiah, V., W. Erwin, Phase I Study of P-cadherin-targeted Radioimmunotherapy with (90)Y-FF-21101 Monoclonal Antibody in Solid Tumors. Clin Cancer Res, 2020. 26(22): p. 5830-5842), I would suggest the authors to discuss the dosimetric outcomes of the 2 studies, comparing data on humans and those obtained on primates to double-check for consistency and reproducibility.
Native speakers
